

Manuscript prepared for Atmos. Meas. Tech.
with version 2015/04/24 7.83 Copernicus papers of the LaTeX class copernicus.cls.
Date: 22 June 2016

# Vertical profiles of the 3D wind velocity retrieved from multiple wind LiDARs performing triple range-height-indicator scans

M. Debnath[1], G. V. Iungo[1], R. Ashton[1], W. A. Brewer[2], A. Choukulkar[2],
R. Delgado[3], J. K. Lundquist[4, 5], W. J. Shaw[6], J. M. Wilczak[2], and D. Wolfe[7]

[1]Wind Fluids and Experiments (WindFluX) Laboratory, Mechanical Engineering Department, The University of Texas at Dallas, Richardson TX
[2]National Oceanic and Atmospheric Administration, Earth Sciences Research Laboratory, Boulder CO
[3]University of Maryland Baltimore County, Baltimore MD
[4]National Renewable Energy Laboratory, Golden CO
[5]Department of Atmospheric and Oceanic Sciences, University of Colorado at Boulder, Boulder CO
[6]Pacific Northwest National Laboratory, Richland WA
[7]Physical Sciences Division, National Oceanic and Atmospheric Administration, Boulder CO

*Correspondence to:* G. V. Iungo (valerio.iungo@utdallas.edu)

**Abstract.** Vertical profiles of the 3D wind velocity are retrieved from triple range-height-indicator (RHI) scans performed with multiple simultaneous scanning Doppler wind lidars. This test is part of the eXperimental Planetary boundary layer Instrumentation Assessment (XPIA) campaign carried out at the Boulder Atmospheric Observatory. The three wind velocity components are retrieved,

then compared with the data acquired through various profiling wind lidars, and high-frequency wind data obtained from sonic anemometers installed on a 300-m meteorological tower. The results show that the magnitude of the horizontal wind velocity and the wind direction obtained from the triple RHI scans are generally retrieved with good accuracy. However, poor accuracy is obtained for the evaluation of the vertical velocity, which is mainly due to its typically smaller magnitude, and

the error propagation connected with the data retrieval procedure and accuracy in the experimental setup.

## 1 Introduction

Wind Light Detection and Ranging (lidar) systems have been employed for wind velocity measurements in different disciplines, such as meteorology (Banta et al., 2002; Calhoun et al., 2006;

Emeis et al., 2007; Horanyi et al., 2015; Vanderwende et al., 2015; Bonin et al., 2015), aeronautic transportation (George and Yang, 2012; Smalikho and Banakh, 2015), wind engineering (Jakobsen et al., 2015) and wind energy (Aitken et al., 2012, 2014; Iungo et al., 2013a; Iungo and Porté-Agel, 2014; Banta et al., 2015; Iungo, 2016). Specifically for wind energy, wind lidars are widely used for characterization of the atmospheric boundary layer (ABL) thanks to their relatively easy deploy-



ment, non-intrusivity, and lower deployment and maintenance costs than for traditional met-towers
(Barthelmie et al., 2010; Schepers et al., 2012).

A Doppler wind lidar allows probing the atmospheric wind field by means of a light beam, which
is backscattered in the atmosphere due to the presence of aerosol. The velocity component along the
light beam direction, denoted as radial or line-of-sight velocity, is evaluated from the Doppler shift

of the backscattered light. Different scanning strategies can be designed to characterize different
properties of the ABL velocity field (Sathe and Mann, 2013; Iungo and Porté-Agel, 2013b; Banta
et al., 2015). The highest spectral resolution of the wind lidar measurements is achievable by max-
imizing the sampling frequency of the lidar and measuring over a fixed direction citepIungo2013a.
3D fixed-point measurements can be performed by retrieving the radial velocity measured simulta-

neously by three or more lidars intersecting at a fixed position (Mikkelsen et al., 2008; Mann et al.,
2009; Carbajo-Fuertes et al., 2014; Berg et al., 2015).

Vertical profiles of the 3D wind velocity within the ABL can be obtained by scanning the lidar
laser beam over a conical path or through the Doppler beam swinging (DBS) technique (Courtney
et al., 2008; Smalikho et al., 2013). These scanning techniques can be leveraged for the character-

ization of the incoming wind of a utility-scale wind turbine (Aitken et al., 2012). However, they
are based on the assumption of a uniform wind field over horizontal planes within the measurement
volume. Therefore, a significant error can be encountered for very heterogeneous flows, such as for
wind turbine wakes (Bingöl et al., 2009) or ABL flows over complex terrain (Lundquist et al., 2015).

Details about the morphology connected with ABL flows can be achieved by sweeping the ele-

vation angle of the lidar, while keeping the azimuthal angle fixed, i.e. performing the range-height
indicator (RHI) scan (Käsler et al., 2010; Hill et al., 2010). The wind velocity field over a volume
including the rotor disc of a utility-scale wind turbine can be measured with intersecting RHI scans
and dual-Doppler lidar retrieval (Newsom et al., 2015). The velocity field of a wind turbine wake
can be characterized over a vertical plane through RHI scans, albeit the continuous adjustment of

the turbine yaw angle complicates the detection of the relative position between the wake and the
measurement plane (Iungo et al., 2013a; Iungo and Porté-Agel, 2013b; Aitken et al., 2014).

Plan position indicator (PPI) scans are performed by varying the azimuthal angle of the lidar laser
beam, while keeping the elevation angle fixed, thus probing a conical surface. PPI scans are highly
suitable for detection and characterization of wind turbine wakes for different wind directions, wake

dynamics and meandering (Iungo et al., 2013a; Aitken et al., 2014; Banta et al., 2015). A series of
consecutive PPI and RHI scans produces a volumetric scan (Banta et al., 2013; Iungo and Porté-Agel,
2014; Banta et al., 2015; Machefaux et al., 2015), which may be useful for a 3D characterization of
the radial velocity within wind turbine wakes.

For this study, four scanning Doppler wind lidars were programmed in order to perform simulta-

neous RHI scans. Various measurement planes are selected in order to determine specific locations
for which two lidars perform co-planar RHI scans, while a third lidar measures over a plane roughly



perpendicular to the one probed by the other two lidars (Fig. 1). With this measurement procedure, at the intersection location of the three lidar measurement planes, a vertical profile of the 3D velocity wind field is retrieved, producing the so-called virtual tower scanning technique. Virtual towers were produced at two separate locations during the experiment. The proposed scanning strategy has been inspired by previous works for which co-planar RHI scans were performed (Hill et al., 2010; Cherukuru et al., 2015). For the first time, at least to the authors' knowledge, the co-planar RHI scan strategy is assessed against other measurement techniques, such as sonic anemometers and wind lidar profilers. Furthermore, in this experiment a third lidar is included in order to perform RHI scans over a plane roughly perpendicular to that of the co-planar RHI scan. As it will be described in the following, this third lidar does not affect accuracy of the velocity components retrieved by the co-planar RHI technique, but it will allow the estimation of the third orthogonal velocity component.

Accuracy of the triple Doppler lidar retrieval from simultaneous intersecting RHI scans is then assessed by comparing the retrieved wind velocity data with the measurements acquired with two profiling wind lidars and sonic anemometers installed on a 300-m met-tower located in proximity of the virtual tower locations (Mikkelsen et al., 2008; Mann et al., 2009; Carbajo-Fuertes et al., 2014).

The remainder of the paper is organized as follows: a description of the instruments used in the experiment is provided in section 2. The data retrieval of the 3D velocity from triple RHI scans is described in section 3, together with the error analysis performed through comparisons with data collected from the lidar profilers and sonic anemometers. Concluding remarks are then reported in section 4.

## 2 Experimental setup and measurement procedures

The eXperimental Planetary boundary layer Instrument Assessment (XPIA) field study was funded by the U.S. Department of Energy within the Atmosphere to electrons (A2e) program to estimate the accuracy and capabilities of various remote sensing techniques for the characterization of complex atmospheric flows in and near wind farms. The XPIA experiment was carried out at the National Oceanic and Atmospheric Administration (NOAA), Boulder Atmospheric Observatory (BAO) near Erie, Colorado for the period March 2 - May 31, 2015.

The field deployment comprised sonic anemometers installed over the BAO met-tower, profiling lidars, radiosonde launches, microwave radiometers, and two scanning Ka-band radars. Moreover, five scanning Doppler wind lidars were deployed to explore novel scanning strategies for the characterization of ABL flows. The triple range-height-indicator (RHI) scan, which is the focus of this paper, is one of the tested scanning strategies. More details about the XPIA campaign can be found in Lundquist et al. (2016b, a).

The BAO met-tower was built in 1977 to investigate the planetary boundary layer (Kaimal and Gaynor, 1983). This 300-m tall tower has three legs spaced 3 m apart and it is instrumented with



temperature and relative humidity sensors at 10 m, 100 m, and 300 m above ground level (AGL), while twelve 3D sonic anemometers CSAT3 by Campbell Scientific were installed at 50 m, 100 m, 150 m, 200 m, 250 m, and 300 m AGL. Six anemometers were installed on booms pointing NW

(334°), which are denoted as NW sonic anemometers, while other six anemometers were installed on SE booms (154°), denoted as SE sonic anemometers. Most of the booms were 4.3 m long, while at the 250 m level the SE boom was 3.3 m long. The sonic anemometers collected data with a sampling frequency of 20 Hz. Sonic anemometer data were tilt-corrected following the method proposed in (Wilczak et al., 2001). The sonic anemometer were calibrated for the XPIA experiment, with mea-

surement resolution (maximum offset error) of 0.1 cm s$^{-1}$ (8 cm s$^{-1}$) for the horizontal velocity and 0.05 cm s$^{-1}$ (4 cm s$^{-1}$) for the vertical velocity McCaffrey et al. (2016).

Two Leosphere/NRG WINDCUBE v1 profiling lidars (denoted as V1) were deployed by CU-Boulder and NCAR's Research Applications Laboratory during XPIA (Aitken et al., 2012; Rhodes and Lundquist, 2013). 3D vertical profiles of the wind velocity were carried out with the Doppler

beam swinging (DBS) technique with an elevation angle from vertical of 28°, and range gates were centered from 40 m to 220 m AGL with steps of 20 m. Similar scans were performed with one Leosphere WINDCUBE Offshore 8.66 profiling lidar, which is denoted as V2. The V2 lidar acquired data at 11 vertical heights (40 m, 50 m, 60 m, 80 m, 100 m, 120 m, 140 m, 150 m, 160 m, 180 m, 200 m). The sampling frequency for the lidar profilers was about 1 Hz. All the lidar profilers

were deployed at the location referred to as lidar supersite and reported in Fig. 1. Its GPS coordinates are reported in Table 1. The profiling lidar data were assessed against sonic anemometer data during XPIA, showing a very good agreement with mean difference of -0.03 m s$^{-1}$ and R$^2$ of 0.97 (Lundquist et al., 2016b, a).

Four scanning Doppler wind lidars were deployed for this experiment. The setup comprises four

Leosphere WINDCUBE 200S (University of Texas at Dallas (UTD), NOAA Dalek1, NOAA Dalek2,

**Table 1.** GPS locations of the four scanning Doppler wind lidars, two virtual towers generated with the triple RHI scans, wind lidar profilers (lidar supersite) and BAO tower.

|  | Longitude | Latitude | Elevation |
|---|---|---|---|
| UTD | W 105°0′3.99″ | N 40°3′2.32″ | 1578 m |
| Dalek1 | W 105°0′55.64″ | N 40°2′51.75″ | 1578 m |
| Dalek2 | W 105°0′20.65″ | N 40°2′43.09″ | 1585 m |
| UMBC | W 105°0′18.90″ | N 40°3′2.56″ | 1577 m |
| Virtual tower 1 | W 105°0′30.82″ | N 40°2′56.73″ | 1578 m |
| Virtual tower 2 | W 105°0′16.77″ | N 40°2′59.58″ | 1578 m |
| BAO tower | W 105°0′13.82″ | N 40°3′0.13″ | 1579 m |
| Lidar supersite | W 105°0′14.36″ | N 40°2′55.72″ | 1580 m |





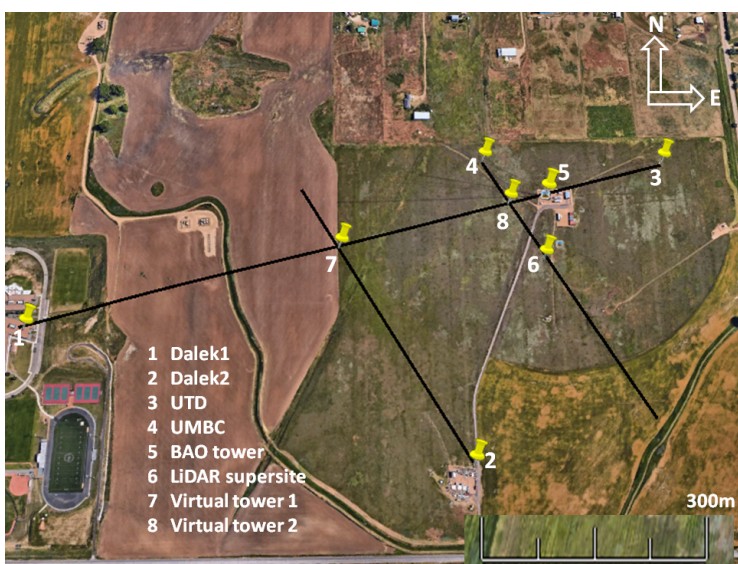

**Figure 1.** Map of the setup for the triple RHI scans performed during the XPIA experiment at BAO. Locations of the four scanning Doppler wind lidars, the two virtual towers, wind lidar profilers (lidar supersite) and BAO tower are reported.

and University of Maryland Baltimore County (UMBC)). Wind measurements were performed by means of an eye-safe laser with a pulse energy of 0.1 mJ and wavelength of 1.54 $\mu$m. Measurements were acquired by using an accumulation time of 0.5 s and gate length of 50 m. Locations of the four scanning Doppler wind lidars are shown in Fig. 1, while their GPS positions are reported in Table 1. Accuracy in the radial velocity of each scanning lidar is always smaller than 0.5 m s$^{-1}$, while the angular resolution of the scanning head is smaller than 0.01°.

All the scanning lidars performed simultaneous RHI scans during the time period 0300-0500 UTC on April 21, 2015. All the lidars used an accumulation time of 500 ms for each line-of-sight position, while different range gates were selected to ensure a good quality of the velocity signals (see Table 2). Ranges of the elevation angles for the RHI scans of the various lidars were selected in order to cover heights between 50 m and 320 m AGL for virtual tower 1, and between 20 m and 90 m for virtual tower 2. For each height of the virtual tower and each lidar, the closest range gate to the considered measurement point is selected for the data retrieval. The maximum horizontal distance of a gate centroid from the respective tower measurement point is 25 m, while the vertical one is always smaller than 10 m. No spatial interpolation of the lidar data was carried out for the data retrieval of the triple RHI scan. Details of the setup for the RHI scans are reported in Table 2. The UTD lidar measured with an azimuthal angle of $\theta = 71.93°$ from North, Dalek1 with $\theta = 251.93°$, UMBC lidar with $\theta = 332°$, and Dalek2 with $\theta = 154°$.



**Table 2.** Parameters of the different scanning lidars for the triple RHI scans.

|        | Azimuthal angle (°) | Elevation angle range (°) | Angular resolution (°) | Gate length (m) |
|--------|---------------------|---------------------------|------------------------|-----------------|
| UTD    | 71.93               | 0-45                      | 1                      | 50              |
| Dalek1 | 251.93              | 0-45                      | 1                      | 50              |
| Dalek2 | 154 and 244         | 0-45                      | 1                      | 50              |
| UMBC   | 332                 | 0-45                      | 1                      | 25              |

**Table 3.** Distance of the four scanning Doppler wind lidars from their respective virtual towers.

|                | Virtual tower 1 (m) | Virtual tower 2 (m) |
|----------------|---------------------|---------------------|
| UTD            | 647                 | 314                 |
| Dalek1         | 626                 | 955                 |
| Dalek2         | 480                 | -                   |
| UMBC           | -                   | 98                  |
| BAO tower      | 415                 | 71                  |
| lidar supersite| 393                 | 136                 |

Intersections of the various RHI measurement planes determine two virtual towers, whose GPS
coordinates are reported in Table 1. Distances of the lidars from the virtual tower locations are re-
ported in Table 3. For virtual tower 1, the UTD lidar covered the measurement range with an average
time period of 13 s, while on average 20 s were required to cover the remaining higher heights and
restart a consecutive scan in raster mode, i.e. in the opposite direction than the previous one. Simi-
larly, Dalek1 requires an average period of 13 s to measure the vertical profile over virtual tower 1
and 19.5 s to restart the next scan. Dalek2 required on average 18 s to measure the vertical profile
and 37 s to restart the next scan. A longer period between consecutive scans was required for Dalek2
due to the scan schedule involving other measurements. Moreover, Dalek2 periodically performed
Planar Position Indicator (PPI) scans with an average scan period of 6 minutes and intervals between
consecutive PPI scans of 12 minutes. Analogous data for virtual tower 2 are reported in Table 4. 3D
velocity profiles at the virtual tower locations were retrieved for time periods for which the three
respective RHI scans overlap.

The lidars were not synchronized, thus different time periods of overlapping were obtained due
to the different delays of the lidar systems. Histograms of the overlapping period for the two virtual
towers are reported in Fig. 2. For virtual tower 1, the overlapping time is generally smaller than
2 s, while for virtual tower 2 all three lidars scanned continuously over the height range, and the
overlapping time has an upper bound limited by the sampling period of Dalek1, which is equal to
3.5 s. The collected lidar data is further post-processed only if the carrier-to-noise ratio of the lidar
data is larger than -17 dB (Carbajo-Fuertes et al., 2014).





**Table 4.** Average sampling period, $t_s$, and time interval between consecutive scans, $t_r$, for the various lidars performing the different virtual towers.

|         | Virtual tower 1 | | Virtual tower 2 | |
| --- | --- | --- | --- | --- |
|         | $t_s$ (s) | $t_r$ (s) | $t_s$ (s) | $t_r$ (s) |
| UTD     | 13 | 19   | 6   | 28 |
| Dalek1  | 13 | 19.5 | 3.5 | 38 |
| Dalek2  | 18 | 37   | -   | -  |
| UMBC    | -  | -    | 21  | 4  |

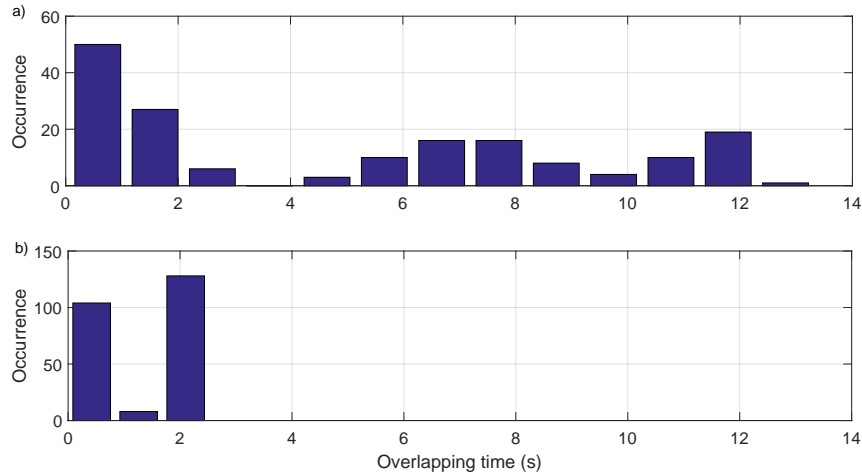

**Figure 2.** Histograms of the overlapping time between the different lidars for the virtual towers: a) virtual tower 1; b) virtual tower 2.

## 3 Retrieval and assessment of 3D wind velocity from triple RHI scans

Data retrieval is described in detail for the virtual tower 1; similar procedures apply to virtual tower 2. For virtual tower 1, the UTD lidar and Dalek1 performed RHI scans over the same vertical plane but with a difference of 180° for the azimuthal angle of their scanning heads (see Fig. 1). Therefore, when the two lidars are set with the same elevation angle, at a given location they will measure a radial velocity with same magnitude and opposite sign. Simultaneously, Dalek2 performed RHI

scans over a plane roughly orthogonal to the one probed by the other two lidars (Dalek1 and UTD). Specifically, the measurement plane of Dalek2 is shifted by an azimuthal angle $\Delta\theta = -7.93°$ (positive is a clockwise shift towards higher azimuthal angles) with respect to the orthogonal plane, while $\Delta\theta = -9.93°$ for virtual tower 2.




Three orthogonal velocity components are retrieved, namely the in-plane horizontal velocity, $U_{in}$,
which lies on the measurement plane of the UTD lidar and Dalek1, the horizontal transversal veloc-
ity, $U_{tr}$, which is orthogonal to $U_{in}$, and the vertical velocity, $W$. These three velocity components
can be evaluated from the radial velocities of the three lidars as follows:

$$
\begin{bmatrix} U_{in} \\ U_{tr} \\ W \end{bmatrix} = \begin{bmatrix} cos(\phi_{UTD}) & 0 & sin(\phi_{UTD}) \\ sin(\Delta\theta)cos(\phi_{D2}) & cos(\Delta\theta)cos(\phi_{D2}) & sin(\phi_{D2}) \\ -cos(\phi_{D1}) & 0 & sin(\phi_{D1}) \end{bmatrix}^{-1} \times \begin{bmatrix} U_r^{UTD} \\ U_r^{D2} \\ U_r^{D1} \end{bmatrix} \tag{1}
$$

where $\phi$ and $U_r$ represent elevation angle and radial velocity of the various lidars, respectively. From
Eq. (1), the three orthogonal velocities can be retrieved directly from the three radial velocities as
follows:

$$
\begin{cases}
U_{in} = \frac{sin(\phi_{D1})U_r^{UTD} - sin(\phi_{UTD})U_r^{D1}}{cos(\phi_{UTD})sin(\phi_{D1}) + sin(\phi_{UTD})cos(\phi_{D1})} \\
U_{tr} = \frac{U_r^{D2}}{cos(\phi_{D2})cos(\Delta\theta)} - \frac{U_r^{UTD}[cos(\phi_{D1})sin(\phi_{D2}) + cos(\phi_{D2})sin(\phi_{D1})sin(\Delta\theta)]}{cos(\phi_{D2})cos(\Delta\theta)[cos(\phi_{UTD})sin(\phi_{D1}) + sin(\phi_{UTD})cos(\phi_{D1})]} \\
\quad - \frac{U_r^{D1}[cos(\phi_{UTD})sin(\phi_{D2}) + cos(\phi_{D2})sin(\phi_{UTD})sin(\Delta\theta)]}{cos(\phi_{D2})cos(\Delta\theta)[cos(\phi_{UTD})sin(\phi_{D1}) + sin(\phi_{UTD})cos(\phi_{D1})]} \\
W = \frac{cos(\phi_{D1})U_r^{UTD} + cos(\phi_{UTD})U_r^{D1}}{cos(\phi_{UTD})sin(\phi_{D1}) + sin(\phi_{UTD})cos(\phi_{D1})}
\end{cases} \tag{2}
$$

The in-plane velocity, $U_{in}$, and the vertical velocity, $W$, are retrieved only from $U_r^{UTD}$ and $U_r^{D1}$,
and are not affected by the measurements carried out with the lidar Dalek2. However, the transversal
velocity, $U_{tr}$, is probed only by the lidar Dalek2, but the retrieval of $U_{tr}$ is a function of the radial
velocities measured by the three lidars.

Accuracy in sensing the 3D velocity field with the triple Doppler lidar technique is dependent
on the setup of the three lidars, thus on the combination of their elevation and azimuthal angles.
The three lines of sight should be set in order to be optimally sensitive to the three orthogonal wind
velocity components (Carbajo-Fuertes et al., 2014). A quantification of the suitability of a triple
Doppler lidar setup for probing the 3D wind velocity field is provided by the $L_2$-norm of the rows
of the matrix reported in Eq. (1) (Simley et al., 2016). Divergence of the row norm from the value 1,
both towards larger or smaller values, indicates an increased error in the retrieval of the respective
wind velocity component. The error analysis related to the lidar setup used for the triple RHI scans is
reported in Table 5 for the two virtual towers and heights. The error in the evaluation of the vertical
velocity, $W$, decreases with increasing height of the virtual tower, which is mainly a consequence of
the increased elevation angles of the lidars, thus of a larger projection of the lidar range gates in the
vertical direction. For the two horizontal velocities, $U_{in}$ and $U_{tr}$, the setup is such to produce a very
slowly increasing error for increased heights.

Various bias errors are considered for the data retrieval of the 3D wind velocity. Corrections of
the position of the lidar scanner heads, azimuth and elevation angles, were estimated with hard
target experiments and GPS measurements, which are not detailed here for the sake of brevity (see
Lundquist et al. (2016a) for details). Bias errors are reported in Table 6 for all the lidars, including



**Table 5.** Error analysis on the retrieval of the 3D wind velocity from triple Doppler lidar measurements as a function of the lidar setup for the various virtual towers and heights.

| Virtual tower 1 | | | |
|---|---|---|---|
| Height (m) | $U_{in}$ | $U_{tr}$ | $W$ |
| 60 | 0.7103 | 1.3949 | 7.5324 |
| 80 | 0.7127 | 1.4015 | 5.6686 |
| 100 | 0.7158 | 1.4101 | 4.5547 |
| 120 | 0.7197 | 1.4205 | 3.8157 |
| 140 | 0.7241 | 1.4327 | 3.2908 |
| 160 | 0.7292 | 1.4466 | 2.8996 |
| 180 | 0.7349 | 1.4621 | 2.5978 |
| 200 | 0.7413 | 1.4794 | 2.3583 |
| 250 | 0.7598 | 1.5292 | 1.9337 |
| 300 | 0.7818 | 1.5884 | 1.6582 |
| Virtual tower 2 | | | |
| Height (m) | $U_{in}$ | $U_{tr}$ | $W$ |
| 40 | 0.79345 | 3.7075 | 8.3921 |
| 60 | 0.7950 | 3.7538 | 5.6258 |
| 80 | 0.7972 | 3.8179 | 4.2518 |
| 100 | 0.7999 | 3.8987 | 3.4345 |

**Table 6.** Bias errors used for the triple Doppler data retrieval.

| | Scanner height (m) | Azimuth (°) | Elevation (°) | los velocity (m s$^{-1}$) |
|---|---|---|---|---|
| UTD | 1.37 | 4.93 | -0.89 | 0.6 |
| Dalek1 | 1.37 | 3.45 | 0.0 | 0.0 |
| Dalek2 | 1.37 | 7.70 | 0.0 | 0.0 |
| UMBC | 1.37 | -40.87 | -0.64 | -0.5 |

bias errors in the radial velocity, which were estimated from fixed vertical velocity measurements
performed over one-day periods. Bias in the radial velocity was due to improper calibration of the
frequency chirp in the laser pulse, which was stable and reproducible in several tests independent of
sonic anemometer comparison, and could simply be subtracted out of the lidar measurements.

Intercomparison of the 3D wind velocity field retrieved from the triple RHI scans with the profiler
wind lidars V1 and V2, and the sonic anemometer data acquired from the BAO met-tower is gener-
ally performed by down-sampling data with higher sampling frequency to the timestamps of the data



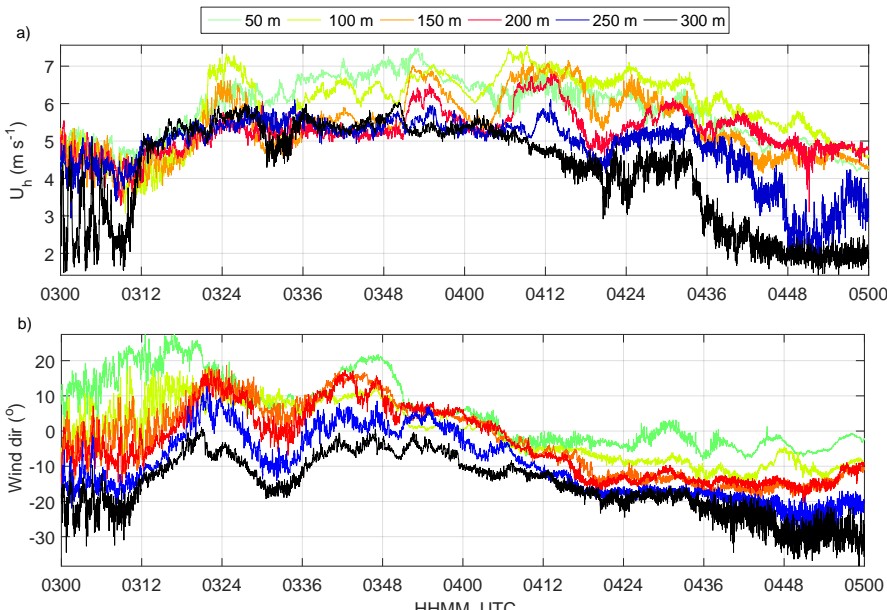

**Figure 3.** Wind velocity measurements obtained from the NW sonic anemometers installed on the met-tower:
a) horizontal velocity; b) wind direction. April 21, 2015, 0300-0500 UTC.

with lower sampling frequency. For instance, the sonic anemometer data acquired with a sampling
frequency of 20 Hz are interpolated to the timestamps of the triple RHI scans by averaging the sonic
anemometer data over the corresponding time period of each lidar data. Similarly, the triple RHI
data is interpolated on the 2 minute averaged data obtained from the lidar profilers V1 and V2.

We note that the sonic anemometers can experience wake effects from the tower for specific wind
directions, i.e. $111° \leq \theta \leq 197°$ for the NW anemometers and $299° \leq \theta \leq 20°$ for the SE anemome-
ters (Lundquist et al., 2016b, a; McCaffrey et al., 2016). For this experiment, wind direction varied
between $330°$ and $20°$, which indicates that the SE anemometers might be affected by wake effects.
Horizontal velocity and wind direction measured by the NW sonic anemometers during the experi-

ment are reported in Fig. 3. Wind speeds were generally low, with an average peak velocity of 5.9
m s$^{-1}$ at about 100 m and average turbulence intensity of 5.6%. The time-averaged Obukhov length
estimated over the entire duration of the experiment form a sonic anemometer installed at a 5-m
height was 4.6 m, thus with a stability parameter of $z/L \approx 1.087$.

Fig. 4 shows the collected radial velocities and retrieved wind velocity components for the period

0300-0500 UTC on April 21, 2015 at virtual tower 1. In Figs. 4a, b, and c, the measured radial
velocities show qualitatively the characteristic sampling period of the three lidars and time intervals





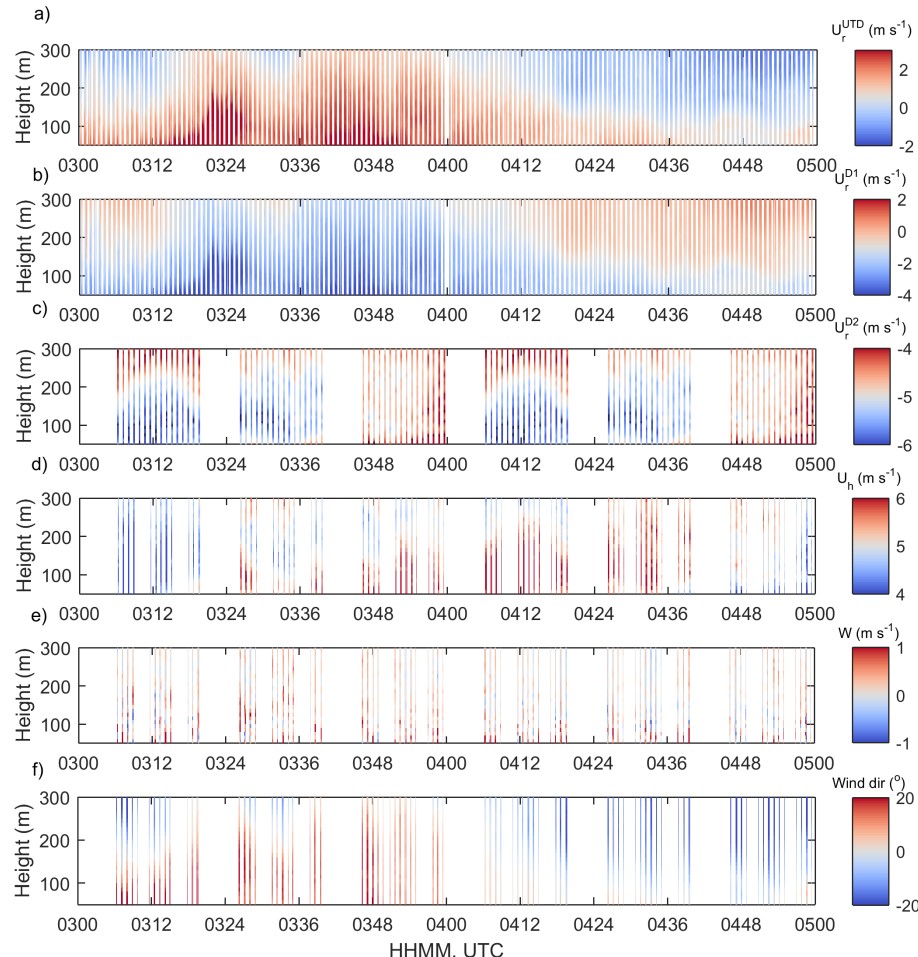

**Figure 4.** Wind velocity measurement for virtual tower 1: a) UTD lidar radial velocity, $U_r^{UTD}$; b) Dalek1 radial velocity, $U_r^{D1}$; c) Dalek2 radial velocity, $U_r^{D2}$; d) horizontal velocity, $U_h$; e) vertical velocity, $W$; f) wind direction.

between consecutive scans. For Dalek2, longer periods with no collected data are observed, which are connected with the time periods when PPI scans were performed.

A detailed assessment of the triple RHI scans with sonic anemometer and lidar profiler data is now presented for virtual tower 1 at a height of 100 m. The radial velocities measured from the three lidars are reported in Fig. 5a. The in-plane and vertical velocities are then retrieved from the radial





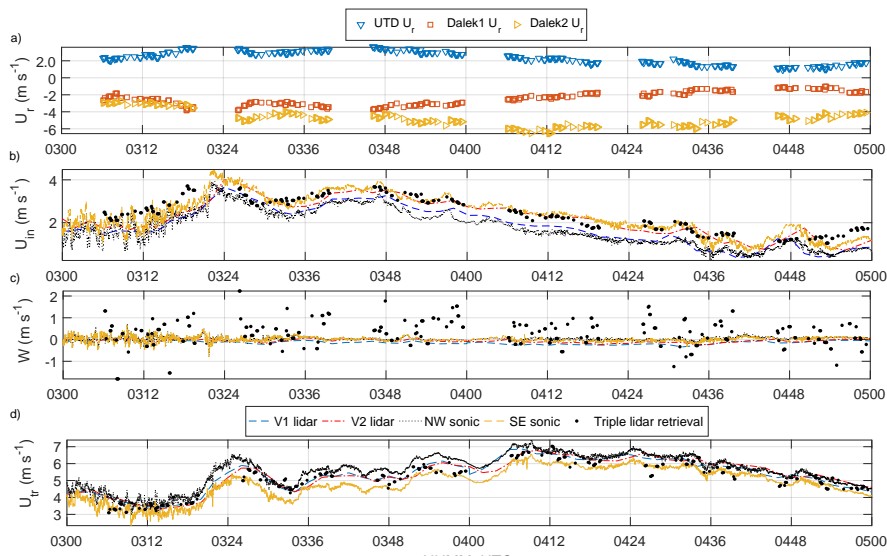

**Figure 5.** 3D velocity retrieved for virtual tower 1 at 100 m height. Assessment of the triple RHI scans with sonic anemometer, and lidar profilers data: a) radial velocities; b) in-plane horizontal velocity, $U_{in}$; c) vertical velocity, $W$; d) transverse horizontal velocity, $U_{tr}$.

velocities $U_r^{UTD}$ and $U_r^{D1}$ as for Eq. (2). As shown in Fig. 5b, $U_{in}$ estimated from the triple RHI scan is in good agreement with that obtained from the other measurement techniques, with a mean square value of the difference equal to 0.09 $m^2s^{-2}$ by comparing to the SE sonic anemometer data,

0.15 $m^2s^{-2}$ relative to the NW anemometer data, 0.18 $m^2s^{-2}$ with the lidar profiler V1, and 0.09 $m^2s^{-2}$ using the V2 lidar profiler and all the respective levels. The estimated difference is the result of the accuracy of the wind lidars, the post-process procedure, the relatively short sampling time, which is consequent to the overlapping time of the different RHI scans (Fig. 2), and the distance between the locations of the virtual tower, lidar profilers and met-tower (Table 3 and Fig. 1).

A larger error is generally encountered for the retrieval of the vertical velocity, $W$ (Fig. 5c). The mean square value of the velocity difference is 0.27 $m^2s^{-2}$ compared with the SE sonic anemometer, 0.24 $m^2s^{-2}$ with the NW anemometer, 0.3 $m^2s^{-2}$ with the lidar profiler V1, and 0.25 $m^2s^{-2}$ with the V2 lidar profiler. This large difference in the measurement of the vertical velocity confirms the estimate of the retrieval error analysis reported in Table 5. Then, by injecting $U_{in}$ and $W$ in Eq. (2),

the transveral velocity $U_{tr}$ is obtained. Fig. 5d, shows that $U_{tr}$ retrieved from the triple RHI scans agrees generally well with the one obtained from the other instruments (mean square value of the difference with respect to the other instruments is 0.15 $m^2s^{-2}$ with the SE sonic anemometer, 0.24



$\text{m}^2\text{s}^{-2}$ with the NW anemometer, 0.17 $\text{m}^2\text{s}^{-2}$ with the lidar profiler V1, and 0.15 $\text{m}^2\text{s}^{-2}$ with the V2 lidar profiler).

Accuracy in the evaluation of the 3D wind velocity from triple RHI scans is assessed through linear regression with respective velocities evaluated from the NW and SE sonic anemometers, and the lidar profilers V1 and V2. From Fig. 6, it is already evident that the two horizontal velocity components, $U_{in}$ and $U_{tr}$, are retrieved with a good accuracy. However, accuracy in the estimate of the vertical velocity, $W$, is very poor. In Fig. 7, slopes and $R^2$ values of the linear regression

are reported for the various instruments and velocity components. Accuracy in the estimate of the in-plane horizontal velocity, $U_{in}$, is generally good, with average slope of 1.01 and $R^2$ of 0.93. A lower agreement with the sonic anemometer data is observed for levels higher than 200 m, due to the low quality of the velocity signals of the sonic anemometers. This is a general feature for all the three velocity components. Regarding the horizontal transversal component, $U_{tr}$, a slightly lower

accuracy is estimated, with an average slope of 0.88 and $R^2$ of 0.81. The retrieval of the vertical velocity is very poor with an average slope of 0.03 and $R^2$ of 0.01.

     Histograms of the error in the retrieval of the 3D velocity from the triple RHI scans, which are obtained by comparing the retrieved data with other instrument data, are reported in Fig. 8. In this figure, in addition to the typical error in the data retrieval, fixed bias errors are observed. Indeed, the

error histograms are generally not symmetric but skewed towards either positive or negative values. These bias errors are typically smaller than 1 m s$^{-1}$, but still noticeable. As mentioned above, the bias errors can also be a consequence of the relatively short sampling time and distance between virtual towers, the lidar profilers and the met-tower.

     Error statistics in the evaluation of the three velocity components from virtual tower 2 are reported

in Table 7, which includes data for heights lower than 90 m. Accuracy in the retrieval of the in-plane horizontal velocity, $U_{in}$, is very good and similar to that obtained for virtual tower 1, while the retrieval of the vertical velocity, $W$, is very poor with an $R^2$ value approximately equal to 0. A lower level of agreement is observed for the retrieval of the transversal horizontal velocity, $U_{tr}$, compared to the results related to virtual tower 1, with and average $R^2$ value of 0.57 and slope of 0.39, which

is due to the different elevation angles of the lidars, as reported in Table 5.

     A strength of the triple RHI scans, compared to other multiple lidar scanning techniques, is the capability of providing vertical profiles of the wind velocity field. By performing time-averages over periods of about 10 minutes, vertical profiles of the horizontal wind speed and direction can be obtained (Figs. 9a and b). For the horizontal wind velocity, generally good agreement is observed

with the time-average velocity profiles obtained from the sonic anemometers installed on the BAO met-tower. A slightly lower velocity is measured by the SE sonic anemometers, which is connected to possible wake effects produced by the met-tower (McCaffrey et al., 2016). For the same reason, some differences are also observed for the wind direction estimated from the triple RHI scans and the one from the sonic anemometers. However, as reported in (McCaffrey et al., 2016), a better estimate





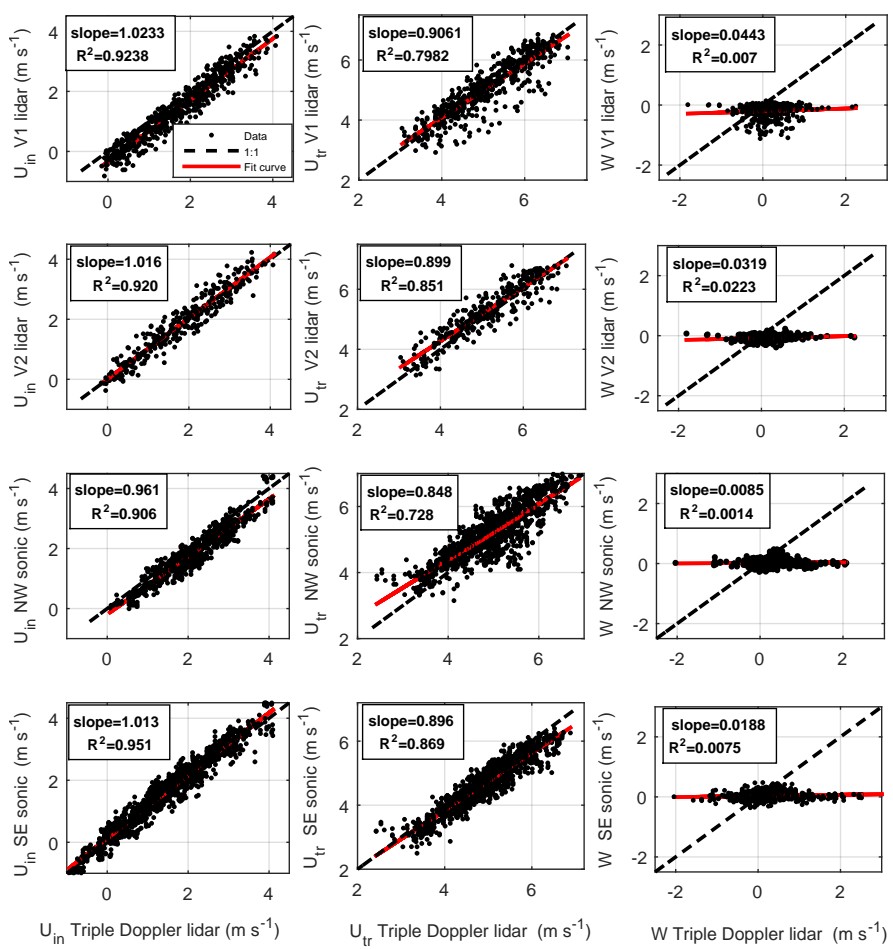

**Figure 6.** Linear regression of the 3D velocity components retrieved from the triple RHI scans with the lidar profilers V1 and V2, and the NW and SE sonic anemometers for virtual tower 1 and all the considered heights.

of the wind direction under waked conditions of the sonic anemometers is obtained by averaging the wind direction measured by the two sonic anemometers at a specific level. By considering this correction procedure, a better agreement between the wind direction estimate by the sonic anemometers and the triple RHI scan is achieved. A noticeable difference is observed with the profiling wind lidars. Regarding the wind direction, very good agreement is observed by comparing the wind data

obtained from the sonic anemometers, especially for heights higher than 150 m. By comparing the





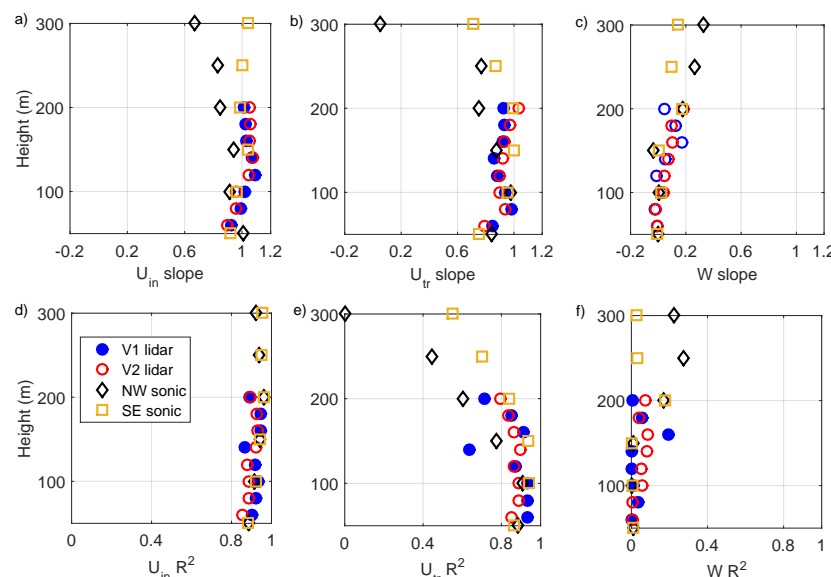

**Figure 7.** Linear regression of the 3D velocity retrieved from the triple RHI scans for virtual tower 1 and compared with the lidar profilers V1 and V2, and the NW and SE sonic anemometers: a) slope of the in-plane horizontal velocity $U_{in}$; b) slope of the transversal horizontal velocity $U_{tr}$; c) slope of the vertical velocity $W$; d) $R^2$ value of the in-plane horizontal velocity $U_{in}$; e) $R^2$ value of the transversal horizontal velocity $U_{tr}$; f) $R^2$ value of the vertical velocity $W$.

wind direction obtained from the triple RHI scans with that obtained from the lidar profilers V1 and V2, a bias error seems to be present between the different measurement techniques. Finally, errors of the mean velocity profiles evaluated as averages over the different heights are reported in Figs. 9c and d for the horizontal velocity and wind direction, respectively. It is evident that errors are generally small.

## 4   Conclusions

Triple range-height-indicator (RHI) scans were performed to retrieve vertical profiles of the 3D wind velocity. This test is part of the eXperimental Planetary boundary layer Instrument Assessment (XPIA) experiment, which was funded by the U.S. Department of Energy and was carried out at the Boulder Atmospheric Observatory in Erie, Colorado, for the period March 2 - May 31, 2015. RHI scans were performed simultaneously with four scanning Doppler wind lidars in order to produce two virtual towers determined by the intersections of their vertical measurement planes. Assessment of the triple Doppler data retrieval has been performed by comparing the triple RHI data with the




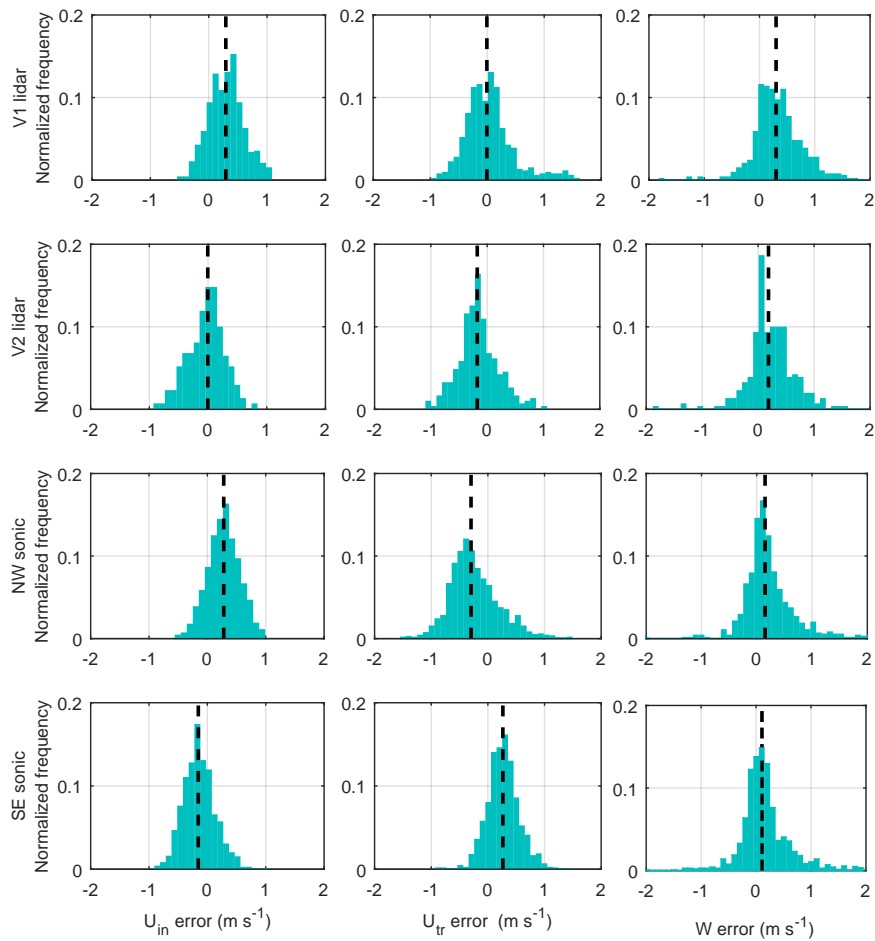

**Figure 8.** Histograms of the velocity difference in the retrieval of the 3D wind velocity from triple RHI scans performed for virtual tower 1 and all the heights, which are obtained through comparison with measurements performed with the lidar profilers V1 and V2, and the NW and SE sonic anemometers. Columns represent different velocity components, rows different instruments. Median is reported with a vertical dashed black line.

wind velocity field measured from two lidar profilers and sonic anemometers installed over the 300-
m tall met-tower present on site.

Intercomparison of the triple RHI data with those obtained from the other instruments has shown that the proposed scanning strategy is highly compelling for producing vertical profiles of the hori-



**Table 7.** Error analysis for the retrieval of the 3D wind velocity from the triple RHI scans at the virtual tower 2. Linear regression with wind measurements performed with the the lidar profilers V1 and V2, and NW and SE sonic anemometers.

| Height (m) | $U_{in}$ $R^2$(slope) | $U_{tr}$ $R^2$(slope) | $W$ $R^2$(slope) |
|---|---|---|---|
| V1 lidar | | | |
| 60 | 0.9422 (1.0292) | 0.4781 (0.4275) | 0.0058 (0.0085) |
| 80 | 0.9424 (0.9902) | 0.5664 (0.3814) | 0.0707 (0.0503) |
| All heights together | 0.941 (1.0105) | 0.5296 (0.3999) | 0.0443(0.0304) |
| V2 lidar | | | |
| 60 | 0.9101 (1.0089) | 0.5665 (0.4541 ) | 0.0089 (0.0091) |
| 80 | 0.9209 (0.9632) | 0.6126 (0.3894) | 0.0298 (0.0226) |
| All heights together | 0.9151 (0.9859) | 0.5917 (0.4149) | 0.0262 (0.0202) |
| NW sonic anemometer | | | |
| 50 | 0.9335 (1.1121) | 0.3744 (0.3698) | -0.0024 (0.0005) |
| SE sonic anemometer | | | |
| 50 | 0.9485 (1.0188) | 0.4691 (0.3808) | 0.0077 (0.0053) |

zontal wind velocity and wind direction. Indeed, very small errors (average correlation of 0.93 and slope 1 for the horizontal velocity, and correlation of 0.8 and slope 0.88 for the wind direction) are

encountered, which are mainly related to the accuracy in the triple lidar setup, relatively short sampling periods, and distance between the virtual towers, lidar profilers and the met-tower. However, low-elevation triple RHI scans are generally not suitable for the characterization of the vertical velocity of the wind field. In case an accurate estimate of the vertical velocity is required, the triple RHI scan setup should be designed with one lidar measuring directly the vertical velocity, while the

other two lidars should have a shift of 90° in the azimuthal angle and the smallest possible elevation angle according to the characteristics of the site and the carrier-to-noise ratio of the lidar signals.

*Acknowledgements.* The authors acknowledge A. J. Clifton for his contribution to the XPIA experiment. This paper was developed based upon funding from the Alliance for Sustainable Energy, LLC, Managing and Operating Contractor for the National Renewable Energy Laboratory for the U.S. Department of Energy.





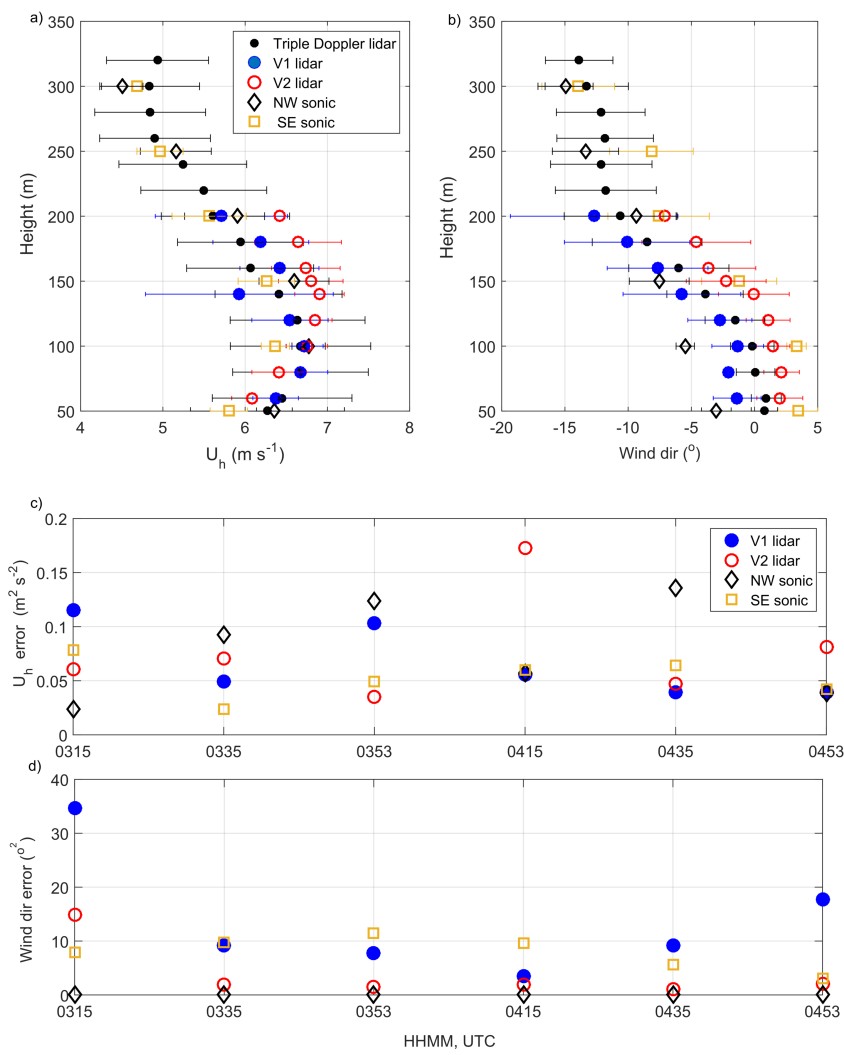

**Figure 9.** Time-averaged vertical velocity profiles and error analysis: a) average in-plane velocity, $U_h$, for the time period 0410-0420 UTC; b) average wind direction for the time period 0410-0420 UTC; c) error in $U_h$ for the different time-averaged vertical profiles; d) error in wind direction for the different time-averaged vertical profiles.

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
