# Peer review of "Vertical profiles of the 3D wind velocity retrieved from multiple wind LiDARs performing triple range-height-indicator scans"

_Atmospheric Measurement Techniques, 2016_

## Referee Comment (RC1) · M. Courtney (Referee) · 16 Jul 2016

General comments The paper describes using scanning lidars to perform 'virtual mast' measurements (scanning along a vertical line) and makes comparisons with co-located physical met mast measurements. Virtual met mast measurements are already, and will increasingly become, useful additions to many field campaigns so the subject is certainly scientifically relevant. The burning scientific questions are; how accurately can we measure means speeds, turbulence and extremes and what are the main limitations? In my opinion, whilst demonstrating one particular technique for virtual met mast measurements, this paper does not contribute very significantly to the relevant scientific issues.

[Figure]

Firstly, the trajectory chosen here is a so-called co-planar RHI (2 scanning lidars facing each other and measuring in a vertical plane) plus a third out of plane scanning lidar. This requires a very specific geometry (two lidars placed on a line intersecting the 'mast' position, here equi-distant from the 'mast') that is easy enough on a test field but very rarely practical in real life measurements where the choice of lidar positions is usually much more restricted. Why was this special case chosen rather than a more typical non co-planar configuration? What are the advantages (above recovering the vertical speed from only 2 lidars) and what are the limitations? Such issues should be covered in the paper.

Secondly, the measurements presented are 2 hours out of a campaign lasting nearly 3 months. Only 6 (six!) discrete values are presented in the error analysis (Figure 9). This can not be a good basis for concluding anything about the accuracy of a measurement technique. Even if these results are considered to show good accuracy, how do we know if these are representative? How will the accuracy vary with wind speed, wind direction, turbulence intensity, wind shear, temperature, other things we haven't thought about. . . . . .? I do not believe that a 2 hour 'snap-shot' is a good scientific basis for concluding anything (good or bad) about measurement accuracy.

Specific comments Purpose of the paper – Please state clearly in the Introduction what is the purpose of the paper – i.e. what is the scientific question you are addressing. I am really not clear whether it is to look at vertical profiles (as the paper title suggests but are 2 hours of wind profiles very interesting?) or to conduct an error analysis (does this make sense with so few samples?). Line 68 'Accuracy . . . is then assessed' – What does 'accuracy' really mean here; difference to the reference sensors (how accurate are these?) or uncertainty? Line 99 ' sonic anemometers were calibrated..' – What does this mean and how was it done? I don't think this should be left to a reference (not available yet anyway) since here we are establishing the accuracy of our central reference instrument. How does 'maximum offset error' relate to uncertainty? Why is instrument resolution given as a calibration result? Line 112 'mean difference of -0.03

m/s' ..' Is this an aggregate of all 3 profiling lidars at all heights? The Rˆ2 is rather low for a direct lidar/mast comparison. Line 120 – 'Accuracy of the radial velocity .. is always smaller than 0.5 m/s' – again what is meant by 'accuracy'? Is this maximum difference? Line 121 – 'angular resolution . . . is smaller than 0.01 deg' – yes, but what about the accuracy? This can be a major source of measurement error but is not really addressed. What about range accuracy (also very relevant for an inclined beam)? Was this checked? Line 124 – 'range gates were selected to ensure a good quality of the velocity signals..' – Please explain what you mean here. Shouldn't you choose the range gates closest to the reference points? Line 130 – 'the vertical distance is always less than 10m' – What magnitude of wind speed error could this introduce? Line 147 – Why was the synchronisation relatively poor? How much difference would better synchronisation have made? Why not make some staring measurements at just one height to give some idea of the extra error introduced by the scanning? Line 196 – 'frequency chirp' – do you mean the AOM frequency shift? Figure 3 – The plot shows wind speeds at low heights (50 and 100m) is significantly higher that at 300m. Is this correct (in which case this should be commented on in the text) or are the signatures mixed up? Line 210 – What is meant by' average peak velocity'? Line 212 – Is the stability at 5m very interesting when the heights seem so de-coupled? Why not use one of the reference heights? Is a time average very useful? Line 226 – 'The estimated difference is the result of the accuracy. . ..' Why is the difference 'estimated'? How can you know that is (really) is the result of the items you list? Maybe re-phrase to 'Possible factors contributing to the observed differences are . . .'. Line 231 – It would be relevant here to also report the differences between the various references. Line 247 – 'lower agreement .. due to the poor quality of the velocity signals of the sonic anemometers.' – This is a very bold statement. How do you know this is true, why is it, and why are you wasting your (and our) time comparing the lidars to something you don't believe is working properly? Line 258 – what other explanations (for the differences) could there be? Figure 6 – The authors claim that the vertical components are well correlated with the sonics and lidars. In my experience, lidars will usually correlate well (and better

than this) to mast instrumentation. It would be interesting to see the lidar profilers correlation with the sonics. Is this better than the 3xRHI? If so, what is the point of the 3xRHI? Generally why have so many references (lidars and sonics) been chosen? Does this help the reader (I think not)? Line 290 – As asked before, why so short a period out of such a long campaign? And why these 2 hours in particular? This is important but should be explained clearly early in the paper.

---

## Author Comment (AC1) · 25 Aug 2016

**Reply to the review by M. Courtney**

We thank the Reviewer for his comments. Our replies are reported below.

1. *The trajectory chosen here is a so-called co-planar RHI (2 scanning lidars facing each other and measuring in a vertical plane) plus a third out of plane scanning lidar. This requires a very specific geometry (two lidars placed on a line intersecting the 'mast' position, here equi-distant from the 'mast') that is easy enough on a test field but very rarely practical in real life measurements where the choice of lidar positions is usually much more restricted. Why was this special case chosen*

*rather than a more typical non co-planar configuration? What are the advantages (above recovering the vertical speed from only 2 lidars) and what are the limitations? Such issues should be covered in the paper.*

**Reply**: The triple and co-planar RHI scans are highly compelling measurement strategies when investigating flows with a prevailing mean wind direction, or vorticity structures and eddies evolving with a specific direction. As mentioned in Sect. 1, in Hill *et al.* 2010, co-planar RHI scans were performed to characterize the vortical motion of eddies generated during mountain-wave events. In Cherukuru *et al.* 2015, co-planar RHI scans were performed to investigate downslope-windstorm-type flows over a plane aligned with the slope of a crater. In Iungo *et al.* 2013, co-planar RHI scans were performed to investigate the wind field over the vertical symmetry plane of a wind turbine wake. In that paper, turbulent statistics of the streamwise and vertical velocities were obtained, together with the corresponding momentum flux. These measurements are highly valuable for wind turbine wake modeling and tuning of turbulence closure models. For this kind of applications, multiple RHI scans allow obtaining multiple measurement points over the plane of interest by using the different range gates of the pulsed lidars, thus achieving small sampling periods. Furthermore, the third lidar enables the retrieval of the three velocity components as a vertical profile. Performing these measurements as consecutive triple fixed-point measurements, i.e. with three lidars setup with a generic arrangement, will allow obtaining only the virtual tower without the remaining 2D velocity measurements over the plane of interest. Obtaining the remaining velocity data over the plane would lead to extremely long, thus unfeasible, sampling periods. The drawback of performing co-planar and triple RHI scans consists in probing the vertical velocity with only two lidars rather than three as for the fixed-point triple scan. Thus, for relatively low elevation angles of the lidar laser beams, a lower accuracy in the vertical velocity might be retrieved. These aspects are now better highlighted in the manuscript.

[Figure]

2. *The measurements presented are 2 hours out of a campaign lasting nearly 3 months. Only 6 (six!) discrete values are presented in the error analysis (Figure 9). This can not be a good basis for concluding anything about the accuracy of a measurement technique. Even if these results are considered to show good accuracy, how do we know if these are representative? How will the accuracy vary with wind speed, wind direction, turbulence intensity, wind shear, temperature, other things we haven't thought about? I do not believe that a 2 hour 'snapshot' is a good scientific basis for concluding anything (good or bad) about measurement accuracy.*
**Reply**: During the XPIA experiment, twelve measurement strategies were tested, and the triple RHI scan is one of these strategies. Approximately one day of measurements was devoted to the triple RHI scan; however, in early Spring aerosol conditions were such to limit the carrier-to-noise ratio, thus data availability. We agree that duration of the presented wind data, which has been selected after a quality control process of the measurements, is relatively short for an assessment study. However, we consider that this assessment against met-tower and two lidar profiler data provides an initial benchmark for the scan capabilities. Important information have been achieved through this study, such as the typical low accuracy in the retrieval of the vertical velocity. It is quite possible that atmospheric variability might influence these results, but the dataset here establishes a first assessment of the method against both in situ meteorological tower data as well as two separate profiling lidars. More comments are now reported in the manuscript.

3. *Purpose of the paper – Please state clearly in the Introduction what is the purpose of the paper – i.e. what is the scientific question you are addressing. I am really not clear whether it is to look at vertical profiles (as the paper title suggests but are 2 hours of wind profiles very interesting?) or to conduct an error analysis (does this make sense with so few samples?).*

**Reply**: As mentioned in Point 1, co-planar and triple RHI scans have been already performed for different experiments and never assessed. This experiment represents the first validation of triple RHI scans against data obtained from sonic anemometers and lidar profilers.

4. *Line 68 Accuracy . . . is then assessed – What does 'accuracy' really mean here; difference to the reference sensors (how accurate are these?) or uncertainty?*
   **Reply**: Accuracy is defined with respect to the reference sensors, such as sonic anemometers and lidar profilers.

5. *Line 99 ' sonic anemometers were calibrated.' – What does this mean and how was it done? I don't think this should be left to a reference (not available yet anyway) since here we are establishing the accuracy of our central reference instrument. How does 'maximum offset error' relate to uncertainty? Why is instrument resolution given as a calibration result?*
   **Reply**: Calibration of the sonic anemometers was performed by the supplier company Campbell Scientific .

6. *Line 112 'mean difference of -0.03 m/s' ..' Is this an aggregate of all 3 profiling lidars at all heights? The RËȨ2 is rather low for a direct lidar/mast comparison.*
   **Reply**: That's correct, the values reported are obtained as ensemble statistics. A slightly lower $R^2$ values might be due to the distance between the location of the lidar profilers, referred to as lidar supersite, the BAO tower and the virtual tower locations, which are all reported in Table 3.

7. *Line 120 – 'Accuracy of the radial velocity .. is always smaller than 0.5 m/s' – again what is meant by 'accuracy'? Is this maximum difference?*
   **Reply**: 0.5 m/s is the accuracy certified by Leosphere for the WindCube 200S, i.e. the maximum difference between the actual LOS wind velocity and the LOS velocity measured by the lidar.

8. *Line 121 – 'angular resolution . . . is smaller than 0.01 deg' – yes, but what about the accuracy? This can be a major source of measurement error but is not really addressed. What about range accuracy (also very relevant for an inclined beam)? Was this checked?*
**Reply**: Accuracy in the laser pointing has been evaluated through hard target tests performed pointing the lidars against the met-tower. These tests allowed estimating the bias errors in azimuthal and elevation angles reported in Table 6. The actual pointing accuracy was estimated to be less than 0.1 deg, while the repeatability, which was estimated using clockwise and counter-clockwise scans to hit a hard target, was 0.01 deg for the azimuth and 0.05 deg for elevation. These details are now added to the manuscript.

9. *Line 124 – 'range gates were selected to ensure a good quality of the velocity signals..' – Please explain what you mean here. Shouldn't you choose the range gates closest to the reference points?*
**Reply**: This part is related to the general setup of the lidars rather than to the selection of the gates for the inter-comparison survey. This sentence has been rephrased: ... while the range gate was 50 m for all the lidars but 25 m for the UMBC lidar (Sect. 2, lines 144-145).

10. *Line 130 – 'the vertical distance is always less than 10m' – What magnitude of wind speed error could this introduce?*
**Reply**: It is difficult providing a firm number for this error, due to the unconventional vertical profiles of the wind field measured during the experiment (see e.g. Figs. 3 and 9a). A certain variability of this error is expected for different wind conditions. However, this error is included in the assessment study.

11. *Line 147 – Why was the synchronisation relatively poor? How much difference would better synchronisation have made? Why not make some staring measurements at just one height to give some idea of the extra error introduced by the*

[Figure]

*scanning?*

**Reply**: Trigger of WindCube 200S is not possible on commercial lidars. However, as explained in the text, data retrieval for the triple RHI is only performed when a certain overlapping among the three RHI scans is obtained.

12. *Line 196 – 'frequency chirp' – do you mean the AOM frequency shift? Figure 3 – The plot shows wind speeds at low heights (50 and 100m) is significantly higher that at 300m. Is this correct (in which case this should be commented on in the text) or are the signatures mixed up?*

    **Reply**: Yes, we mean the AOM frequency shift. This LOS velocity offset introduced by this frequency shift was estimated by using three independent techniques: 1) weighted averages of the signals obtained through hard target experiments; 2) four hour averages of vertical staring measurements; 3) comparison against sonic anemometers. All the three tests provided the same offset in the radial velocity. This offset was periodically checked during the experiment, which resulted to be quite constant. The velocity signals are correct and most probably being the result of stable conditions and the footprint of the surrounding topography.

13. *Line 210 – What is meant by' average peak velocity'?*

    **Reply**: We meant the maximum velocity over the height of the mean velocity profiles (Sect. 3, lines 230-231).

14. *Line 212 – Is the stability at 5m very interesting when the heights seem so decoupled? Why not use one of the reference heights? Is a time average very useful?*

    **Reply**: Characterization of atmospheric stability has been done through the eddy-covariance method. Therefore, a better evaluation of the contribution of mechanically produced turbulence and buoyancy driven turbulence is obtained from sonic anemometer measurements carried out within the surface layer.

15. *Line 226 – 'The estimated difference is the result of the accuracy. . ..' Why is the difference 'estimated'? How can you know that is (really) is the result of the items you list? Maybe re-phrase to 'Possible factors contributing to the observed differences are...*
    **Reply**: In the text is reported that the estimated difference is the result of lidar accuracy, post-process, relatively short sampling time and overlapping among the various lidars, and accuracy in lidar pointing. Therefore, the values provided represent an overall accuracy.

16. *Line 231 – It would be relevant here to also report the differences between the various references.*
    **Reply**: In the following table, differences among all the instruments are reported.

| Instruments | $U_{in}$ | $U_{tr}$ | $W$ |
|---|---|---|---|
| V2 lidar - V1 lidar | 0.16 | 0.03 | 0.01 |
| V2 lidar - SE sonic | 0.03 | 0.20 | 0.02 |
| V2 lidar - NW sonic | 0.21 | 0.10 | 0.02 |
| V1 lidar - SE sonic | 0.18 | 0.28 | 0.04 |
| V1 lidar - NW sonic | 0.05 | 0.07 | 0.03 |
| SE sonic - NW sonic | 0.19 | 0.24 | 0.01 |
| V2 lidar - 3RHI | 0.09 | 0.15 | 0.25 |
| V1 lidar - 3RHI | 0.18 | 0.17 | 0.30 |
| NW sonic - 3RHI | 0.15 | 0.24 | 0.24 |
| SE sonic - 3RHI | 0.09 | 0.15 | 0.27 |

Similar differences are estimated for $U_{in}$ and $U_{tr}$ among all the instruments. This new table clearly shows that the triple RHI scan produces larger differences with respect to the other instruments in the retrieval of the vertical velocity, with an average slope of the linear regression with other instruments of 0.01 and $R^2 = 0.01$.

17. *Line 247 – 'lower agreement .. due to the poor quality of the velocity signals of the sonic anemometers.' – This is a very bold statement. How do you know this is true, why is it, and why are you wasting your (and our) time comparing the lidars to something you don't believe is working properly?*
    **Reply**: We agree with the Reviewer that this sentence is misleading. It is now reported: A lower agreement with the sonic anemometer data is observed for levels higher than 200m, which might be due to the larger velocity fluctuations measured at higher levels (see Fig. 3).

18. *Line 258 – what other explanations (for the differences) could there be?*
    **Reply**: As mentioned, differences might be due to short sampling time, separation distance between the virtual towers, lidar profilers and met-tower.

19. *Figure 6 – The authors claim that the vertical components are not well correlated with the sonics and lidars. In my experience, lidars will usually correlate well (and better than this) to mast instrumentation. It would be interesting to see the lidar profilers correlation with the sonics. Is this better than the 3xRHI? If so, what is the point of the 3xRHI? Generally why have so many references (lidars and sonics) been chosen? Does this help the reader (I think not)?*
    **Reply**: By performing linear regression between sonic and profiling lidar data we obtain an average for $U_{in}$ slope=0.86 and $R^2 = 0.94$; for $U_{tr}$ slope=0.85 and $R^2 = 0.85$; for $W$ slope=0.46 and $R^2 = 0.35$ (Sect. 3, lines 265-267). This indicates that comparable accuracy for the horizontal velocities is achieved among the different instruments. The main drawback of the co-planar and triple RHI scans remains a poor accuracy in the retrieval of the vertical velocity. A limited correlations between the different measurement techniques might be due to the separation distances among the different instruments, which are reported in Table 3. However, the advantage of the triple RHI scans consists in obtaining wind data over a preferred vertical plane with a relatively short sampling time. Furthermore, virtual towers can probe volumes not accessible from met-tower and with

lower costs.

20. *Line 290 – As asked before, why so short a period out of such a long campaign? And why these 2 hours in particular? This is important but should be explained clearly early in the paper.*
**Reply**: During the XPIA experiment, twelve measurement strategies were tested, and the triple RHI scan is one of these strategies. Approximately one day of measurements was devoted to the triple RHI scan; however, in early Spring aerosol conditions were such to limit the carrier-to-noise ratio, thus data availability. Two hours of data is the result of the quality control process of the data and time management for the various experiments during the XPIA campaign.

Please also note the supplement to this comment:
http://www.atmos-meas-tech-discuss.net/amt-2016-164/amt-2016-164-AC1-supplement.pdf

**Supplement:**

[revised manuscript text omitted]

Co-planar and triple RHI scans are highly compelling measurement strategies when investigating flows with a prevailing mean wind direction, such as for wind turbine wakes, or vorticity structures and eddies evolving with a specific direction. Co-planar RHI scans were performed to characterize the vortical motion of eddies generated during mountain-wave events (Hill et al., 2010). In

- 65 Cherukuru et al. (2015), co-planar RHI scans were performed to investigate down-slope-windstormtype flows over a plane aligned with the slope of a crater. Co-planar RHI scans were also performed to investigate the wind field over the vertical symmetry plane of a wind turbine wake (Iungo et al., 2013a). In that paper turbulent statistics of the streamwise and vertical velocities were obtained, together with the corresponding momentum flux. These measurements are highly valuable for wind
- 70 turbine wake modeling and tuning of turbulence closure models. For this kind of applications, coplanar and triple RHI scans allow obtaining multiple measurement points over the vertical plane of interest by using the different range gates of the pulsed lidars, thus achieving small sampling periods. Furthermore, the third lidar enables the retrieval of the three velocity components as a vertical profile at the intersection line among the three RHI planes. Performing these measurements as
- 75 consecutive triple fixed-point measurements, i.e. with three lidars setup with a generic arrangement, would lead to extremely long, thus unfeasible, sampling periods. For the first time, at least to the authors' knowledge, the multiple RHI scan strategy is assessed against other measurement techniques, such as sonic anemometers and wind lidar profilers. Furthermore, in this experiment a third lidar is included in order to perform RHI scans over a plane roughly perpendicular to that of the co-planar
- 80 RHI scan. As it will be described in the following, this third lidar does not affect accuracy of the velocity components retrieved by the co-planar RHI technique, but it will allow the estimation of the third orthogonal velocity component.

[revised manuscript text omitted]

 RHI scans, wind lidar profilers (lidar supersite) and BAO tower.

(Lundquist et al., 2016b). The slightly lower correlation between sonic anemometers and lidar pro-filers might be due to the separation distance between the met-tower and the location of the lidar profilers (Table 3).

Four scanning Doppler wind lidars were deployed for this experiment. The setup comprises four Leosphere WINDCUBE 200S (University of Texas at Dallas (UTD), NOAA Dalek1, NOAA Dalek2, and University of Maryland Baltimore County (UMBC)). Wind measurements were performed by

- 135 means of an eye-safe laser with a pulse energy of 0.1 mJ and wavelength of 1.54  $\mu$ m. Measurements were acquired by using an accumulation time of 0.5 s and gate length of 50 m. Locations of the four scanning Doppler wind lidars are shown in Fig. 1, while their GPS positions are reported in Table 1. Accuracy in the radial velocity of each scanning lidar is always smaller than 0.5 m s-1, while the angular resolution of the scanning head is smaller than 0.01°. Accuracy in the laser pointing was
- 140 evaluated through hard target tests by pointing the lidars against the met-tower. These experiments allowed estimating the bias errors in azimuthal and elevation angles (Table 6). The actual pointing accuracy was estimated to be less than  $0.1^{\circ}$ , while repeatability, which was estimated through consecutive clock- and counter clock-wise scans, was estimated to be  $0.01^{\circ}$  for the azimuthal angle and  $0.05^{\circ}$  for the elevation angle.
- 145 During the XPIA experiment, twelve lidar scanning strategies were tested, and the triple RHI scan was performed for approximately one day. However, the poor local aerosol conditions occurring in early Spring led to a relatively low carrier-to-noise ratio of the lidar velocity signals, thus to a limited data availability. Although this dataset represents the first assessment of the scanning strategy under examination, the relatively short sampling period (0300-0500 UTC on April 21, 2015) of this
- 150 experiment does not allow estimating effects of wind and atmospheric conditions on the accuracy of the triple RHI technique.

[revised manuscript text omitted]

---

## Referee Comment (RC2) · Anonymous Referee #1 · 15 Sep 2016

This paper concern the use of lidars as "virtual tower" for measuring wind. The amount of data used is very low and there are problems with the way the experiment has been planned. It is hard to point out what new knowledge is generated by the paper.

The experiment is not planned very logically. All scanners use elevation angles from 0-45 degrees while have very different distances to the virtual towers ranging from 98 to 955 m. This means that a lot of measurements are waisted, i.e. that the overlap in measurements is quite poor. Furthermore, the scanners are not synchronized which means that they interrogate the same intersecting volume at the virtual mast at different times. The implication of both these issues is that the calculation of the wind vector becomes more uncertain than it needed to be. Also the azimuth angles are not ideal.

[Figure]

They should have been closer to 120 degrees apart to reduce uncertainties on the horizontal wind, or, if reduced uncertainty in the vertical component is wanted, have one close to the foot of the virtual tower and the two others at roughly 90 degrees apart. For the virtual mast 2 one instrument is in fact close to the base of the virtual mast (8 in fig 1) but the two other lidars are at 180 degrees to each other, virtually the worst configuration one could choose.

The comparisons in figure 6 are unimpressive which is probably due to the bad setup of the experiment as mentioned above. Slopes between the sonic are up to 15% off for the horizontal components and much more for the vertical.

---

## Author Comment (AC2) · 9 Oct 2016

We thank the Reviewer for her/his comments. Our replies are reported below.

1. *...The amount of data used is very low and there are problems with the way the experiment has been planned.*
   **Reply**: During the XPIA experiment, twelve measurement strategies were tested, and the triple RHI scan is one of these strategies. Approximately one day of measurements was devoted to the triple RHI scan; however, in early Spring aerosol conditions were such to limit the carrier-to-noise ratio, thus data availability. We agree that duration of the presented wind data, which has been selected after

a quality control process of the measurements, is relatively short for an assessment study. However, we consider that this assessment against met-tower and two lidar profiler data provides an initial benchmark for the scan capabilities. Important information have been achieved through this study, such as the typical low accuracy in the retrieval of the vertical velocity. More comments are now reported in the manuscript.

2. *It is hard to point out what new knowledge is generated by the paper.*
   **Reply**: Co-planar and triple RHI scans have been already performed for different experiments (e.g. Hill *et al.* 2010 for mountain-wave events; Cherukuru *et al.* 2015 for down-slope-windstorm-type flows; Iungo *et al.* 2013 for wind turbine wakes) and never assessed. This experiment represents the first validation of triple RHI scans against data obtained from sonic anemometers and lidar profilers.

3. *The experiment is not planned very logically. All scanners use elevation angles from 0-45 degrees while have very different distances to the virtual towers ranging from 98 to 955 m. This means that a lot of measurements are wasted, i.e. that the overlap in measurements is quite poor. Furthermore, the scanners are not synchronized which means that they interrogate the same intersecting volume at the virtual mast at different times. The implication of both these issues is that the calculation of the wind vector becomes more uncertain than it needed to be. Also the azimuth angles are not ideal. They should have been closer to 120 degrees apart to reduce uncertainties on the horizontal wind, or, if reduced uncertainty in the vertical component is wanted, have one close to the foot of the virtual tower and the two others at roughly 90 degrees apart. For the virtual mast 2 one instrument is in fact close to the base of the virtual mast (8 in fig 1) but the two other lidars are at 180 degrees to each other, virtually the worst configuration one could choose.*
   **Reply**: The aim of this manuscript is not performing triple Doppler measurements

for optimal setups, which can be rarely performed during real experimental campaigns. The aim of this study is rather to assess accuracy of triple RHI scans for the three velocity components under sub-optimal scanning conditions. The errors consequent to the setup of the azimuthal and elevation angles of the three lidars is systematically documented. In Table 3 expected errors in the retrieval of the three velocity components, for the two virtual towers and different heights are reported.

Regarding the overlapping time among the three lidars, the lidars as used do not provide any synchronization through a master supervisory computer. Therefore, as might happen for the majority of field experiments according with the existing lidar technology, instruments were programmed while a certain delay might occur among the various lidars. The overlapping time is statistically documented in Figure 2. We believe that, as long as a significant overlapping time is ensured among the various lidars and data retrieval is only performed for simultaneous data, there is no specific concern on the accuracy of the 3D data retrieval. However, we agree that the scanning strategy is not very efficient due to a significant rejection of non-simultaneous data.

Regarding the setup of the lidars, multiple RHI scans allow obtaining multiple measurement points over the plane of interest by using the different range gates of the pulsed lidars, thus achieving small sampling periods. Furthermore, the third lidar enables the retrieval of the three velocity components as a vertical profile. Performing these measurements as consecutive triple fixed-point measurements, i.e. with three lidars setup as suggested by the Reviewer, will allow obtaining only one virtual tower without the remaining 2D velocity measurements over the plane of interest. Obtaining the remaining velocity data over the plane would lead to extremely long, thus unfeasible, sampling periods. The drawback of performing co-planar and triple RHI scans consists in probing the vertical velocity with only two lidars rather than three as for the fixed-point triple scan. Thus, for

relatively low elevation angles of the lidar laser beams, a lower accuracy in the vertical velocity might be retrieved. These aspects are now better highlighted in the manuscript.

4. *The comparisons in figure 6 are unimpressive which is probably due to the bad setup of the experiment as mentioned above. Slopes between the sonic are up to 15% off for the horizontal components and much more for the vertical.*
**Reply**: The goal of this paper was not to demonstrate the accuracy of the wind lidar retrieval using the absolute best possible setup configuration, but rather to show the errors when using setups that have to meet multiple objectives and with siting limitations, leading to sub-optimal geometries for most of the individual wind retrievals. It is reported in the text that the estimated difference is the result of lidar accuracy, post-process, relatively short sampling time, accuracy in lidar pointing, configuration setup and separation distance among the virtual towers, lidar profilers and met-tower.